# Exposure to Environmental Toxins: Potential Implications for Stroke Risk via the Gut– and Lung–Brain Axis

**DOI:** 10.3390/cells13100803

**Published:** 2024-05-08

**Authors:** Alexandria Ruggles, Corinne Benakis

**Affiliations:** Institute for Stroke and Dementia Research, University Hospital, LMU Munich, 81337 Munich, Germany; alexandria.ruggles@med.uni-muenchen.de

**Keywords:** environmental toxins, pollutants, gut-brain axis, lung-brain axis, microbiome, brain injury, stroke, neuroinflammation

## Abstract

Recent evidence indicates that exposure to environmental toxins, both short-term and long-term, can increase the risk of developing neurological disorders, including neurodegenerative diseases (i.e., Alzheimer’s disease and other dementias) and acute brain injury (i.e., stroke). For stroke, the latest systematic analysis revealed that exposure to ambient particulate matter is the second most frequent stroke risk after high blood pressure. However, preclinical and clinical stroke investigations on the deleterious consequences of environmental pollutants are scarce. This review examines recent evidence of how environmental toxins, absorbed along the digestive tract or inhaled through the lungs, affect the host cellular response. We particularly address the consequences of environmental toxins on the immune response and the microbiome at the gut and lung barrier sites. Additionally, this review highlights findings showing the potential contribution of environmental toxins to an increased risk of stroke. A better understanding of the biological mechanisms underlying exposure to environmental toxins has the potential to mitigate stroke risk and other neurological disorders.

## 1. Introduction

Over the past two centuries, society has undergone profound changes driven by the rise of urbanization and industrialization. Advances in society have introduced various hazards directly impacting human health [1]. Among these hazards are biological and chemical pollutants, extreme climate conditions, gas emissions, and heavy metal accumulation, all of which have been linked to an increased risk of various health conditions, not only respiratory complications, but also endocrine disruptions, cancer, cardiovascular, and neurological disorders [2,3,4,5]. Globally, stroke is a highly prevalent neurological disease and a main contributor to long-term disabilities, remaining the second leading cause of death [6]. Importantly, stroke is no longer a disease of just the elderly; over the past two decades, the prevalence and incidence rates of stroke in people under the age of seventy have increased [7]. This phenomenon has been attributed in part to increased risk factors, including poor diet, smoking habits, and chronic exposure to stressors. Additionally, emerging evidence suggests that environmental toxins, including household air pollution, ambient particulate matter, and exposure to heavy metals, contribute to the global stroke burden [8,9]. In fact, the most recent systematic analysis shows that among stroke risk factors, ambient particulate matter exposure is second only to high blood pressure in terms of prevalence [6].

Human exposure to environmental toxins can occur through three primary routes: dietary intake, inhalation, and dermal absorption [1]. Short and long-term exposure to toxins has shown bioaccumulation in different tissues, including the gut, lungs, and central nervous system [10]. In consequence, these toxins can interfere with cellular processes, inducing neurotoxicity and altering the host immune response [11]. Indeed, experimental studies have demonstrated that exposure to a range of environmental toxins can disrupt host homeostasis by impairing the cellular response of the immune system, particularly at barrier sites such as the gut and lungs [12,13,14]. This disruption is further reinforced by recent findings that environmental contaminants also significantly alter the gut microbiome’s composition and function [12,15], suggesting exposure to such toxins has an impact on biological interfaces. Because recent studies highlighted the critical role of the gut–brain axis, in neurological disorders [16] and given that the primary route of exposure to environmental toxins is through food consumption and inhalation, the lung and intestinal microbiota could be significant intermediaries in the influence of these chemical pollutants on neurological disorders.

Despite growing evidence linking environmental toxins to stroke risk factors, the field lacks a thorough mechanistic understanding of the precise pathways through which these toxins influence stroke pathobiology. Research on this topic is limited, leaving a gap in our knowledge of the long-term effects of toxin exposure and the synergistic impact of multiple pollutants on brain diseases. Additionally, socioeconomic status, geographical region, sex difference, and pre-existing conditions that may affect individual susceptibility to these toxins are not well characterized, further complicating risk assessment and mitigation strategies. Bridging these gaps is crucial for developing targeted interventions and refining policy regulations to reduce stroke incidence linked to environmental neurotoxins. Therefore, this review aims to examine the biological mechanisms associated with environmental toxins and deleterious consequences on the host. To do so, it explores the impact of these toxins on the lungs and gut, especially delving into the cellular pathways and microbiome-related changes, and further emphasizing their potential contribution to an increased risk of stroke onset and its outcomes. This narrative review aims to raise awareness of the importance of the gut- and lung-brain axis linked to exposure to environmental toxins and stroke risks, which we believe is relevant for other neurological and neuroinflammatory diseases.

## 2. Environmental Toxins and Stroke Risk

Environmental toxicants are either airborne pollutants or chemical toxins that have been shown to cause systemic inflammation and ultimately becoming neurotoxic [17]. These environmental toxins, grouped under the name of xenobiotics, are nearly impossible to evade in today’s industrialized world and include particulate matter, dioxins, persistent organic pollutants (POPs), and polycyclic aromatic hydrocarbons (PAHs) [18]. Chronic interactions with these substances can incrementally impact human health, leading to the emergence of disease through different exposure pathways. Importantly, exposure to environmental toxins has been shown to increase the occurrence, risk, and development of neurological disorders, including Alzheimer’s disease and dementia, and be a possible risk factor for Parkinson’s disease, and it impacts neuronal development [19,20]. The neuroinflammatory response is a key common component of the pathobiology of these neurological diseases [21,22]. Neuroinflammation is not only observed from repeated or chronic exposure periods but can be induced even after a single insult [23], demonstrating the need to monitor toxin exposure and prolongation to minimize potential threats to human health. Although observational and prospective studies have offered valuable insights into the correlation between exposure to environmental toxins and the onset of various neurological and neuroinflammatory disorders and neurodegenerative diseases [19,24], there is a lack of research elucidating the mechanisms and pathways underlying the role of environmental toxins in neurological conditions such as stroke.

In this section, we discuss pertinent environmental toxins and their metabolic pathways that lead to host toxicity, particularly in the context of stroke. The categories of toxins reviewed include (1) persistent organic pollutants (POPs), (2) per- and polyfluorinated substances (PFASs), (3) air pollutants, and (4) micro- and nanoplastics. Table 1 summarizes experimental studies that demonstrated the biological consequences of environmental toxins on the host cellular response and, when available, the implications for stroke.

### 2.1. Persistent Organic Pollutants

Persistent organic pollutants (POPs) represent a broad class of toxins abundant in the environment and formed as byproducts from industrial or chemical processes, such as waste incineration, vehicle exhaust, and cigarette smoke [50,51,52,53]. Due to their physical properties, these pollutants resist natural degradation processes and persist in the environment. Common POPs include organochlorine pesticides (OCPs), polychlorinated biphenyls (PCBs), polycyclic aromatic hydrocarbons (PAHs), dioxins, and dibenzofurans [50]. These persistent pollutants are ubiquitous in atmospheric, terrestrial, and aquatic environments [52]. Due to their lipophilic properties, POPs accumulate in lipid-rich tissues and organs, including the liver, blood, brain, adipose tissue, and kidney, and bioaccumulate in mammals, humans, fish, and invertebrates [54,55].

#### 2.1.1. Polycyclic Aromatic Hydrocarbons

Organic pollutants characterized by two or more fused benzene rings are classified as PAHs. Biodegradation or detoxification of PAHs by the host involves several enzymatic pathways, including the binding of xenobiotics to the aryl hydrocarbon receptor (AhR) and subsequent expression of P450 (CYP) genes [56,57]. Most of the metabolic products of xenobiotics are excreted from the body, while a significant amount fails to be detoxified and is retained in the body. Indeed, CYP-mediated biotransformation of various POPs can result in toxic compounds that either elicit a mutagenic or genotoxic effect [58,59,60]. For instance, the highly toxic environmental chemical TCDD (2,3,7,8-tetrachlorodibenzo-p-dioxin) has been shown to alter several cellular and tissue responses, leading to neurotoxicity, immunotoxicity, reproductive toxicity, and the formation of tumors [60]. Mechanistically, inhibition of AhR activity, using AhR-deficient mice, prevented the carcinogenic effect of the PAHs: TCDD and benzo(a)pyrene (BaP) [61,62], highlighting the functional role of AhR in mediating deleterious effects of environmental toxins.

Epidemiological studies have shown that PAHs, including BaP, have been associated with hypertension [63] and cardiovascular diseases [64], both known to increase ischemic stroke risks [65]. Regarding the association between stroke and exposure to PAHs, Rahman et al. found that higher levels of specific urinary PAHs were positively correlated with increased odds of stroke onset in men and women [66]. In experimental studies, BaP exposure has been associated with high blood pressure and vasoconstriction (Table 1). Indeed, following both acute and chronic BaP exposure in animal models, BaP increased blood pressure, exceeding 20% of baseline values [65,66]. Moreover, chronic treatment with BaP administered by oral gavage in mice susceptible to atherosclerosis (ApoE-deficient mice) increased the size of atherosclerotic plaques and a local inflammatory response characterized by the infiltration of T lymphocytes and macrophages [67]. Exposure to BaP has a direct effect on brain function since BaP and its metabolites can accumulate in lipid-rich tissues such as the brain [68] and have been shown to impair the blood–brain barrier function [69]. Tanaka et al. demonstrated that exposure to PAHs altered stroke outcomes in mice after intranasal exposure, which induced edema in the brain and impaired locomotor functions in rats [70]. These findings suggest that despite the liver’s primary role in detoxifying xenobiotics [71], there exists a tissue-specific vascular response, potentially highlighting the link between certain PAHs and stroke pathophysiology. The complex interplay between the increased risk of cardiovascular complications linked to PAH exposure and stroke occurrence and outcomes underscores the necessity for deeper insights into how environmental toxin exposure heightens the risk of stroke onset, mortality, and morbidity.

#### 2.1.2. Polychlorinated Biphenyls

Despite the implementation of new regulations to ban the production of some PCBs in 1978 [72], the lasting repercussions of industrial manufacturing continue to persist and have lasting effects on human health. Even decades after the ban of PCBs, Raffetti et al. identified correlations between the fasting blood serum levels of several PCB compounds and the health status of middle-aged individuals residing in a heavily polluted area with a history of PCB production. Significantly, specific PCB congeners, such as 138, 153, and 180, which are considered to be mid- and highly chlorinated and are more resistant to being metabolized, were found to be associated with the onset of hypertension based on fasting blood serum levels [73]. Elevated serum levels of PCBs have also been positively associated with increased blood pressure [74] and an increased risk for ischemic stroke in both men and women [75]. Moreover, as shown in Table 1, passive dietary consumption of PCBs has also been demonstrated to increase the risk of stroke onset, specifically in women [31]. Mechanistically, PCBs, which are ligands of AhR, activate the nuclear transcription factor nuclear factor- κB (NF-κB) pathway, inducing oxidative stress, production of inflammatory cytokines, and upregulation of the expression of inflammatory genes [76,77]. This indicates the role of PCBs in inducing a pro-inflammatory response.

### 2.2. Per- and Polyfluorinated Substances

The production of PFASs began just less than a century ago [78]. Due to their structural and chemical properties, these chemicals are highly resistant to degradation and persist in the environment for extended periods of time. PFAS have been extensively utilized in household and industrial products, including non-stick cookware, medical devices, insulating packaging material, cosmetics, and textiles [79]. The broad utilization of PFAS across various sectors has resulted in widespread accumulation in both the environment and animals, raising concerns regarding their potential impact and health hazards for individuals [78,80]. Ingestion and inhalation are the two primary routes of human exposure to PFAS through dietary consumption, mainly from drinking water [81,82]. It has been reported that the average half-life in human blood serum for some of these substances is up to 8.5 years [83]. Several studies have shown exposure to PFAS is positively associated with an increased risk for hypertension [84]. Despite elevated levels of cholesterol and hypertension observed in the previous studies, which are known risk factors for cardiovascular disease and stroke [85], there have been conflicting results regarding whether exposure to PFAS increases the risk of stroke onset. Simpson et al. found a modest association with stroke incidence in individuals living near a PFAS chemical plant [86], while Feng et al. demonstrated that higher PFAS levels increased the risk for cardiovascular disease and stroke, particularly in males [87]. Another study showed that individuals with higher serum levels of PFAS exhibited dyslipidemia, yet no significant association between stroke and myocardial infarction [88].

Experimental research indicates that exposure to members of the PFAS family induces immunotoxicity by activating signaling pathways, including the NF-kB pathway. This activation triggers the generation of reactive oxygen species (ROS) and facilitates the upregulation of cytokine production [89]. Moreover, tissue inflammation after exposure to PFAS is mediated by several signaling pathways such as peroxisome proliferator-activated receptor α (PPARα) and Janus kinase 2/signal transducer and activator of transcription 3 [90,91]. Although these signaling pathways are activated following experimental stroke [90,91], the exact mechanistic pathways linking PFAS exposure to influencing the risk for stroke remain elusive. Additionally, a new hypothesis has emerged that exposure to PFAS may target the cardiovascular system by activating platelet aggregation and adhesion, promoting thrombosis [92]. A deeper understanding of the health hazards associated with PFAS exposure and the underlying mechanistic pathways could provide insights into how these compounds contribute to stroke risk.

### 2.3. Air Pollutants

The World Health Organization (WHO) reports that a significant portion of premature deaths linked to outdoor air pollution is due to cardiovascular diseases and ischemic stroke (37%), and respiratory infections (23%) (https://www.who.int/news-room/fact-sheets/detail/ambient-(outdoor)-air-quality-and-health; accessed on 22 January 2024). More than a hundred hazardous air pollutants originating from activities including oil and gas production, battery manufacturing, iron and steel forging, plastic manufacturing, and wood preserving are monitored by The Environmental Protection Agency (EPA) monitors (https://www.epa.gov/haps/area-sources-urban-air-toxics; accessed on 22 January 2024). Both short-term and long-term exposure to ambient air pollution are now recognized as a modifiable risk factor contributing to the onset and mortality of ischemic stroke [93]. While everyone is affected by air pollution, people living in urban regions are exposed to a higher concentration of air pollutants compared to those living in rural areas [94]. Consequently, urban residents face an elevated risk for the development of health complications as a result of prolonged exposure and concentration.

According to the National Institute of Health (NIH), air pollution is a mixture of hazardous compounds, which include PAHs, ozone, particulate matter, noxious gases, nitrogen oxides, and sulfur oxides (https://www.niehs.nih.gov/health/topics/agents/air-pollution; accessed on 22 January 2024). More specifically, particulate matter detected in air pollution is characterized by its aerodynamic diameter. Particulate matter with a diameter of ≤10 µm (PM10) is considered coarse, ≤2.5 µm (PM2.5) is considered fine, and ≤0.1 µm (PM0.1) is considered ultrafine [95,96]. Mechanistically, when particulate matter is inhaled, it can accumulate along the respiratory tract. PM10 typically accumulates in the nasal cavity and trachea, whereas PM2.5 can penetrate deeper into the lungs, primarily targeting the bronchioles. The finest particles, those smaller than 2.5 µm, target the alveoli and can even diffuse into the bloodstream [97]. Particulate matter can enter the circulation and can traffic to the brain, acting as a pro-inflammatory stimulus and triggering chronic inflammation [98]. This sustained brain inflammation has been characterized as a consequence of continuous particulate matter exposure inducing ROS production, microglial activation, neuroinflammation, and neuronal damage [23,99,100,101].

In animal models, passive exposure to air pollution induced significant alterations in the lung epithelium, characterized by increased mucus secretion and morphological changes in the pulmonary cell lining. These changes manifested as an augmented presence of microvilli and a reduction in the population of ciliated cells within the lung tissue [102]. In a controlled laboratory setting, individuals without pre-existing health conditions who were exposed to a mixture of common air pollutants, including sulfur dioxide, sulfate aerosols, and ozone, demonstrated an increased risk of developing pulmonary dysfunction, including a decrease in forced expiratory volume and vital capacity [103]. In an epidemiological study investigating the association between PM2.5 exposure and respiratory and cardiovascular complications, findings revealed that women were more likely to be hospitalized for such complications compared to men when exposed to PM2.5 [104].

Numerous epidemiological studies have shown that exposure to PM2.5 and PM10 is associated with a higher ischemic stroke incidence rate and mortality compared to studies investigating ultrafine ambient pollutants [93,105,106,107]. Szyszkowicz et al. observed a correlation between low-level exposure to ambient air pollution and an increased number of visits to the hospital for hypertension [108], which may be an indicator of higher stroke risk for those individuals exposed to ambient air pollution. These findings highlight that air contaminants may compromise an already vulnerable region of the lungs, potentially increasing the risk of lung infections, hospitalizations associated with lung complications, and post-stroke pneumonia, and stroke outcomes should be further explored.

### 2.4. Nano- and Microplastics

The use of plastic globally has continued to increase at a staggering rate, reaching nearly 500 million tons annually. Despite policy efforts and heightened awareness of methods to recycle and dispose of plastics, less than 10% is recycled, and a quarter of the plastic is inadequately disposed of (https://www.oecd.org/environment/plastic-pollution-is-growing-relentlessly-as-waste-management-and-recycling-fall-short.htm; accessed on 14 March 2024). Improperly disposed plastic can persist in the environment and be degraded and broken into fragmented pieces known as microplastics (particles less than 5 mm) and nanoplastics (less than 1 nm) [109,110]. Abundant in the environment, these particles can be both inhaled and ingested as they have been detected in soil, air, and drinking water [111,112]. Not only are these particles accumulating in the environment, but they are also found in several human samples, including stool, blood, and breastmilk [113,114,115]. Many of the most recent studies have investigated the effects of microplastics on animals living in aquatic environments [116,117]. However, there are relatively few studies focusing on the effects in humans. In mice, chronic exposure to micro- and nanoplastics led to the accumulation of particles in the gut [29] and perturbed gut homeostasis. In an experimental model, analysis of the gastrointestinal tract revealed that post-exposure, there was a reduction in overall microbial diversity and compromised gut architecture, characterized by a reduction in the mucosal wall lining and the absence of villi along the gastrointestinal tract [15]. In humans, a recent study showed that individuals with microplastics detected in the coronary arteries were at heightened risk for developing cardiovascular complications and development of stroke [118]. These findings suggest that exposure to micro- and nanoplastics accumulates in the host and influences diseases of the gut, heart, and brain.

## 3. Exposure to Environmental Toxins at Barrier Sites

Barrier sites, including the lungs, gut, and skin, act as physical barriers and immune barriers to maintain host homeostasis and protect from external threats [119,120,121]. If compromised by environmental toxins, these barrier sites have been shown to contribute to brain diseases, specifically regarding the lung–brain axis and gut–brain axis. Cellular mechanisms linking environmental toxins, barrier sites, and stroke pathobiology are discussed in the next sections.

The respiratory tract is lined with an epithelial layer consisting primarily of pseudostratified ciliated columnar cells, which function as a barrier against foreign particles. The cells lined with cilia and mucosal cells initiate a defense mechanism by generating mucus and trapping foreign particles and pathogens [122]. Exposure to environmental toxins such as particulate matter increases ROS production, compromising the epithelial barrier integrity and creating an environment where bacteria are more likely to invade [123], demonstrating the vulnerability of the barrier site. Short-term exposure to PM10 reduced the tight junction protein occludin in human and rat alveolar epithelial cells [124]. In contrast, exposure to PCB126, a dioxin-like compound, increased the expression of the tight junction proteins occludin and claudin [32]. Jang et al. found that ozone exposure also significantly impacts the structural integrity of the lung’s epithelial layer and immune response in the lungs. After exposure to ozone, the researchers observed initial epithelial cell death followed by an increased proliferation of epithelial cells coupled with an increase in neutrophils [125]. In wild-type mice, it was shown that acute exposure to ozone compromised the epithelial lining, increased epithelial damage in the lungs, and increased permeability in ST2-deficient mice [39]. Overall, these findings highlight the complex interaction between environmental toxins and the integrity of the epithelial barrier and immune regulation. While different environmental pollutants can have contrasting effects on epithelial barrier function and immune regulation, exposure may have implications for respiratory health and disease susceptibility.

In the gut, the digestive tract serves as an interface between the environment and the host, acting as a direct entry sight for pathogenic invaders, including bacteria, pathogens, and contaminated food components [126]. As a defense mechanism to prevent bacteria and pathogens from invading the tissue, specialized intestinal epithelial cells known as goblet cells produce mucus, forming a mucosal layer [127,128]. It has been shown that even within the first 24 h post-stroke, intestinal permeability is compromised, contributing to inflammation and bacterial translocation to secondary sites. The impaired permeability of the gut has been directly correlated with an increased severity of stroke outcomes [129]. Environmental toxins can also lead to altered permeability of the gut lining. A 3D model of human intestinal samples revealed that exposure to particulate matter at a concentration of 500 μg/cm^2^ for up to two weeks reduced the levels of zonula occludens protein 1 (ZO−1) and claudin-1 [36]. Mice acutely and chronically exposed to high concentrations of ambient PM2.5 increased the permeability of the gut barrier by reducing the expression of the tight junction protein, ZO-1, at acute and chronic time points [130]. Histological analysis of the gastric epithelium in mice deficient in the tight junction protein occludin did not compromise the structure of the epithelium but rather promoted chronic inflammation [131]. The subsequent inflammatory response in the gut activated by the upregulation of IL-17-producing T cells can increase gut permeability by disrupting the tight junction barrier [132]. One explanation for the impaired architecture of the tight junctions is that cells, as a stress mechanism in response to exposure to particulate matter, will generate ROS, which disrupt tight junction formation and increase epithelial cell permeability [35].

### 3.1. Lung–Brain Axis and Environmental Toxins

Recent findings suggest there is a bidirectional communication pathway involving pulmonary microbiota, their metabolites in the lungs, the central and autonomic nervous system, and the hypothalamic–pituitary–adrenal (HPA) axis, which are collectively known as the lung–brain axis [133]. While still an emerging concept in understanding the functional role of the lung–brain axis in health and disease, the lung–brain axis has been demonstrated to play an integral role in the pathology of neurological diseases. In a recent study, Hosang et al. manipulated the pulmonary microbiome using a cocktail of antibiotics, which in turn influenced the pathogenesis of experimental autoimmune encephalomyelitis (EAE), a rat model of multiple sclerosis [134]. Further evidence supports the idea that environmental toxins can alter the lung–brain axis and may influence the likelihood of stroke onset and post-stroke outcomes. Tanaka et al. demonstrated that intranasal exposure to urban aerosols in mice can affect post-stroke recovery by augmenting neuroinflammation, activating brain resident microglial cells, and impairing motor function post-stroke [70].

Among the various complications experienced by patients following a stroke, urinary infections, and post-stroke pneumonia are the most prevalent [135]. Post-ischemic stroke not only causes site-specific damage but has been shown to impact secondary organs such as the lungs in experimental stroke models [136]. Stroke onset activates an immune cascade and consequently leads to immunosuppression in the periphery, increasing the risk of developing infections, including stroke-associated pneumonia and the patients’ 30-day mortality rate [137,138]. Exposure to air pollution has also been linked to an increased risk of bacterial lung infections, in particular bacterial pneumonia [139]. The association between stroke and lung infection highlights the need for more research unveiling the lung–brain axis (Figure 1).

#### 3.1.1. Environmental Toxins and Immune Response in the Lungs

The lungs are one of the first organs to interact with inhaled particulate matter. Exposure to particulate matter has been shown to impair immune cell functions, modify the immune response, and disturb lung architecture. PM10 can penetrate deeper into the lungs, targeting the lung bronchioles and resulting in remodeling of lung tissue and the upregulation of immune cells [140]. Damage to lung architecture and altered immune response elevate the likelihood of lung diseases such as chronic obstructive pulmonary disease (COPD), chronic bronchitis, and respiratory tract infections due to resulting lung impairment and dysfunction post-exposure to environmental toxins [43,141,142].

Importantly, studies have shown that particulate matter can translocate to secondary organs and elicit a systemic effect. Upon inhalation, particulate matter first enters the nasal cavity and can then translocate into the brain through multiple routes, including the olfactory nerve, compromised blood–brain barrier, or damaged nasal epithelium [143]. Chronic insults from inhaled particulate matter have revealed that aging is associated with a decline in the phagocytic capacity of macrophages and accumulation of particulate matter within the lung-associated lymph nodes, ultimately altering the lung-associated lymph nodes’ follicular architecture [144]. Similarly, Samary et al. demonstrated in rats that ischemic stroke resulted in changes in the respiratory environment, impairing the phagocytic capacity of macrophages, upregulating inflammatory markers including TNF-α and IL-6 levels in the lungs, and inducing lung edema [145]. While it has been shown that exposure to particulate matter can alter the tissue and immune architecture, even in the absence of lung damage, exposure to particulate matter can influence coagulation factors and fibrinogen, bridging the connection between inhaled particulate matter and cardiovascular diseases [146]. Impaired lung function has been demonstrated to result in a higher incidence of ischemic stroke [147]. Together, this highlights the intricate connection between exposure to particulate matter and the crosstalk between the lungs and the cardiovascular system, which may influence the onset of disease.

#### 3.1.2. Effect of Environmental Toxins on the Lung Microbiome

The pulmonary microbiome should be considered when determining the effects of environmental toxins. While it was once thought that the lungs were sterile [148], emerging evidence suggests that manipulating the pulmonary microbiota can activate immune cells in the brain and influence neurological disease outcomes [134]. Variation in bacterial taxa has been mapped along the respiratory tract from the oral cavity through the trachea and to the lungs [149]. Despite the limited number of studies investigating pulmonary microbiota, a recent study demonstrated that inhalation exposure to PM2.5 led to acute inflammation in the bronchioles, increased the number of goblet cells, and reduced the abundance of proteobacteria in the lungs [42]. Potential consequences of exposure to environmental toxins disturb mediators within the lung–brain axis. We suggest that future investigations should invest in understanding the role of the pulmonary microbiota influenced by environmental factors in stroke pathogenesis.

### 3.2. Gut–Brain Axis and Environmental Toxins

The gut–brain axis is a bidirectional communication pathway that signals information between the gut and the brain via the autonomic and enteric nervous system, the hypothalamic–pituitary axis, the vagal nerve, and systemic circulation [150,151]. Humans are exposed to millions of foreign substances referred to as xenobiotics directly and passively. Upon exposure, the body metabolizes and eliminates these substances, with toxic xenobiotics being transformed into less harmful derivatives before being excreted through stool, urine, perspiration, and bile [152]. Xenobiotics are also metabolized by the resident gut microbiota, which can influence bioaccumulation and toxicity [153]. However, accumulation of metabolites in the body from xenobiotics, when not excreted or eliminated, can pose significant health risks by disrupting cellular homeostasis and causing tissue damage [57]. Moreover, oral exposure to environmental toxins will not only accumulate in the digestive tract but can also enter the bloodstream through enterohepatic circulation. Consequently, this can lead to the bioaccumulation of toxic derivatives in secondary organs such as the brain [154].

Trillions of microorganisms reside within the intestinal tract, comprised of bacteria, fungi, protozoans, archaea, and viruses [155]. Metagenomic analysis of the human microbiota without underlying conditions revealed that the phyla change along the digestion tract; however, it is dominated primarily by commensal bacteria: Bacteroidetes, Firmicutes, Proteobacteria, and Actinobacteria phyla [156]. Several factors shape the composition and function of the microbiota, including diet, aging, and the environment [157,158,159]. In particular, the gut microbiota plays an integral role in training and educating the immune system and facilitating tolerance mechanisms [160]. Signaling from the brain can modulate the immune response in the gut, intestinal permeability, and bacterial–host interactions [161]. Recent studies have revealed exposure to environmental toxins shifts the relative abundance of bacteria and alters microbial profiles in animal models. Dietary exposure to heavy metals and PCBs has been shown to reduce the abundance of Proteobacteria while simultaneously increasing the presence of Bacteroidetes [25,26,30]. Chronic exposure to PAHs (BaP and TCDD) also shifts the bacterial ratios, increasing the relative abundance of Bacteroidetes while reducing the abundance of Firmicutes [12,162]. Interestingly, long-term inhalation exposure to PM2.5 altered the ratio of bacteria in mice, marked by a significant reduction in Akkermensia and Firmicutes, and significantly increased the Bacteroidetes [130], demonstrating the impact of inhaled air pollutants on the gut microbiota. Collectively, these experimental findings highlight the impact of environmental toxins on the gut microbiota, yet the impact of alterations of the composition in the context of neurological diseases such as stroke risk and outcomes remain unknown.

#### Exposure to Environmental Toxins and Stroke Risk

Stroke onset triggers an immune response in both the brain and the periphery. It has been established that the occlusion of a cerebral artery causes shear stress in the brain, altering the blood–brain barrier permeability and limiting the amount of glucose and oxygen to the brain tissue, leading to neuronal cell death [163]. This acute response triggers an immune cascade signaling to resident immune cells to activate the innate and adaptive immune response [164]. Inflammation after stroke extends beyond the site of injury and propagates a systemic effect, greatly impacting secondary organs in the periphery [165]. Immune cells have been shown to traffic from peripheral organs, such as the gut and brain, post-stroke to mediate inflammation [166,167]. Because neuroinflammation plays an essential role in the pathobiology of stroke, the effect of environmental toxins on the immune system warrants further investigation.

Several preclinical and clinical studies have demonstrated stroke induces dysbiosis of the gut microbiota, which can influence the severity of stroke outcomes [168,169,170,171]. Singh et al. found that in an experimental stroke model, the expansion of Bacteroidetes is associated with a proinflammatory response marked by the increased expression of IFN-γ+ and IL-17 cytokines produced by T helper cells [168]. Moreover, prior to stroke onset, mice treated with Ampicillin shifted the microbial profile, leading to the expansion of Proteobacteria and Lactobacilliales and a reduction in Bacteroidetes, which were deemed to be neuroprotective by reducing the lesion size after stroke [172]. These changes in the microbiota composition induced an expansion of regulatory T cells in the gut and down-regulation of IL-17 γδT cells, which prevented their trafficking into the brain and thereby prevented the formation of the primary ischemic damage [166]. Together, these studies reveal the intricate relationship between the host microbiota, the gut immune system, and stroke pathobiology. Because environmental toxin exposure induces alterations in the microbiome’s relative abundance, inducing a proinflammatory environment, this complex interplay may influence the immune response and, subsequently, impact stroke outcomes. However, further research is needed to understand the detailed mechanisms and implications of these interactions and how exposure to xenobiotics can influence stroke outcomes.

Inhalation of particulate matter can directly impact the lungs, but it can also indirectly induce inflammation in the gut [34]. Experimental models have shown that inhalation of particulate matter alters the bacterial ratio, as previously mentioned [130], but it also has an impact on the immune cell reservoir in the gut. The authors demonstrated that exposure to air pollution in dense urban regions for two weeks upregulated the expression of inflammatory cytokine production in the colon [34]. These findings highlight that environmental toxins exert a systemic effect at barrier sites, which can disrupt immune cell reservoirs in the periphery. Understanding how environmental toxins shape the immune compartments in peripheral organs and potentially influence the outcomes of stroke could further reveal the complexity of these interactions and improve our understanding of the impact of environmental factors on health outcomes.

## 4. Conclusions and Outlook

Reducing the incidence of strokes requires a comprehensive approach, including addressing exposure to environmental toxins. Prevention requires the implementation of new laws and regulations targeting exposure limits and the production of hazardous material to minimize exposure. Concurrently, detection strategies, on the other hand, provide a measure of an individual’s risk. Measuring metabolite concentrations in urine samples can be used to monitor toxin exposure [173]. Whereas prevention and detection strategies have long-term implications, immediate measures are required to counter-effect the deleterious consequences of environmental toxin exposure for stroke.

To date, there are limited studies understanding how environmental toxins act to mediate their potent effects contributing to stroke development. In this review, we examined numerous clinical, experimental, and epidemiological studies that have demonstrated that exposure to toxins via ingestion and inhalation disrupts cellular homeostasis, in particular the immune system and microbiome, and exerts a systemic effect on the gut, lungs, and brain. In addition, we highlighted how environmental toxins could increase the risk of stroke or exacerbate stroke outcomes.

By understanding the mechanisms of action related to environmental toxin exposure on the gut and lung–brain axis, we could limit the impact of environmental toxins-induced toxicity and associated brain diseases. To develop new therapeutic or preventive strategies, we need future research to address the systemic impact (gut and lungs) of environmental toxins and their contribution to stroke onset. For instance, dietary interventions rich in fiber and polyphenols have been demonstrated to play a role in reducing the bioavailability of xenobiotics by absorbing and minimizing their uptake [174,175]. Characterization of the gut and lung microbiomes associated with immune changes may be another approach to mitigate the effects of environmental toxin exposure. This would lead to new therapeutic strategies targeting microbiome–host interactions to counteract the deleterious effects of environmental pollutant exposure for brain diseases.

In conclusion, this review addresses how environmental toxins induce a systemic effect that may play a critical role in stroke pathogenesis. To our knowledge, there are limited studies that address the impact of environmental toxins in the context of stroke pathogenesis. While this review article provides insights into the role of environmental factors as risk factors for stroke, several gaps in the literature remain. Firstly, the precise mechanisms through which environmental toxins contribute to stroke pathogenesis remain unclear. Secondly, the exploration of sex differences in susceptibility to environmental toxins and their subsequent impact on stroke risk remains largely unexplored. Furthermore, there is a need for research to elucidate the systemic effects of environmental toxin exposure on stroke outcomes. Addressing these gaps will be crucial for advancing a more comprehensive understanding of environmental influences on stroke pathogenesis.

## Figures and Tables

**Figure 1 cells-13-00803-f001:**
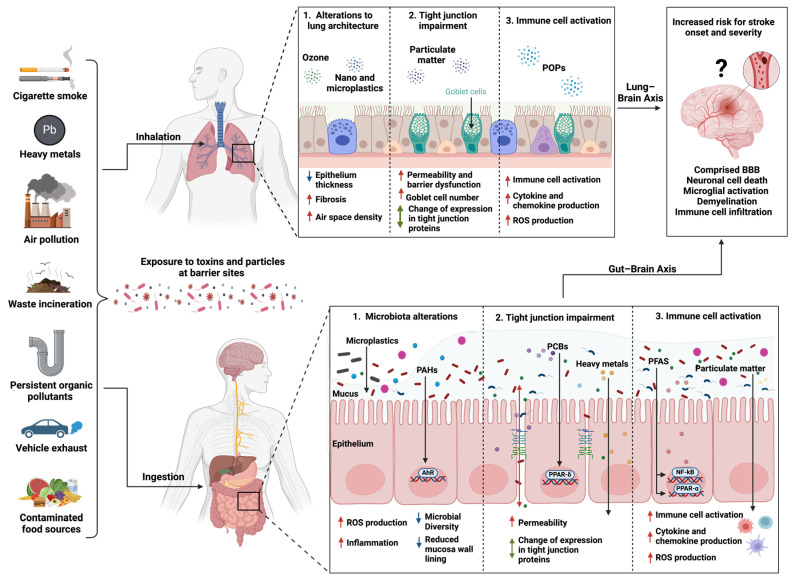
Mechanisms of environmental toxins inhaled or ingested potentially leading to increased susceptibility to stroke and severity. Environmental toxins originate from several sources, including industrial processes and as byproducts from incomplete combustion. These toxins can enter the environment and contaminate sources such as food, water, and soil and can also be abundant in the air, increasing human exposure via inhalation and ingestion as two primary routes. Once exposed, these toxins can target the primary organs, gut and lungs, and induce perturbation of the microbiota composition (dysbiosis), a pro-inflammatory response leading to disruption of tissue architecture, immune cell activation, and increased permeability at epithelial barrier sites, which may increase the exposure risk to harmful bacteria and pathogens. Inhaled or ingested xenobiotics are metabolized and bind to their respective receptors, upregulating ligand-activated transcription factors and inflammatory signaling pathways (AhR, PPAR, and NF-κB). Alterations at the gut and lung barriers can indirectly affect brain function. Metabolites generated from the metabolism of xenobiotics can traffic to the brain and may directly increase the risk for stroke by compromising the blood–brain barrier, activating brain resident immune cells and immune cells trafficking from the periphery, and causing cell death. AhR, aryl hydrocarbon receptor; BBB, blood–brain barrier; NF-κB, nuclear factor-κB; PAHs, polycyclic aromatic hydrocarbons; PCBs, polychlorinated biphenyls; PET, polyethylene terephthalate; PFASs, per- and polyfluoroalkyl substances; POPs, persistent organic pollutants; PPAR-α and -δ, peroxisome proliferator-activated receptor-alpha and -delta; ROS, reactive oxygen species; TNF-α, tumor necrosis factor-alpha; Tregs, regulatory T cells, created with Biorender.com.

**Table 1 cells-13-00803-t001:** Studies linking the effect of environmental toxins on the gut, lungs, and circulatory system and their consequences for stroke risk.

Environmental Toxins	Compound/ Substance	Model	Dosage	Exposure Method and Duration	Effect	References
Gut						
Heavy Metal	Cadmium	Sprague–Dawley rats	5 mg/kg	Gastric infusion daily for 30 days	↓ Expression of the tight junction protein ZO-1 ↑ Expression of TNF-α, and IL-6 in the gut	[25]
Heavy Metals	Cadmium/Arsenic	C57BL/6 mice	50 ppm	Dietary exposure through drinking water for 2 weeks	↓ Alpha diversity of the microbiota	[26]
Heavy Metal	Arsenic	Participants in cohort study	<50 μg/L	20-year exposure window	Higher incidence of stroke	[27]
Microplastics and nanoplastics	Micro/nanoplastics	C57/B6 mice	0.2 and 2 mg/kg	28 days	High dose led to morphological changes along the GI including the followings: -Absence of villi -Damaged crypts ↓ Mucosa wall lining ↓ Tight junction expression. ↓ Microbial diversity and ratio of bacteria	[15]
Microplastics	Polyethylene terephthalate (PET)	In vitro simulation	0.166 g/intake	72 h	Shift in the relative abundance of bacteria in the upper region of the colon. Specifically, ↑ Desulfobacterota and ↓ Proteobacteria over time. ↓ Viable bacterial count	[28]
Microplastics	PS-MPs	Male mice	0.1mg/day of 5 or 20 μm PS-MPs	28 days via oral gavage	Accumulation of PS-MPs in the gut, liver, and kidney. ↑ Inflammation and lipid accumulation in the liver	[29]
PAH	BaP	C57BL/6 mice	10 mL/kg of BW	28 days oral exposure	↑ Inflammation in ileal segments -Altered relative abundance of fecal and mucosa associated microbiota ↓ Lactobacillus	[12]
PCBs	PCB153, PCB138, and PCB180	C57BL/6 mice	150 µmol/kg	2 days	↓ Proteobacteria	[30]
PCBs	Prospective population based Follow up	Middle-aged and elderly women	Passive dietary exposure	12 years of follow-up	Positive association observed between stroke risk and PCB exposure.	[31]
PCB	PCB 126	Ldlr−/− mice	1 μmol/kg of PCB 126	14-week atherogenic diet and exposed to PCB 126 (1 μmol/kg) at weeks 2 and 4	↑ Expression of TNF-α, IL-6, and interleukin IL-18 in the jejunum ↑ Tight junction proteins: Occludin and claudin in the colon ↓ Expression of PPAR-δ in the colon ↓ Abundance of Clostridiales, Bifidobacterium, Lactobacillus, Ruminococcus, and Oscillospira. ↑ Abundance of Akkermansia. ↓ Alpha diversity in cecum contents	[32]
PCB	PCB153	C57BL/6 mice	300 μmol/kg	1× per day for 2-days	↑ Expression of TNF-α and IL-6 in the intestinal epithelial cells of the small intestines -Activates NF-κB pathway -Induces DNA damage	[33]
Particulate matter	Urban particulate matter	C57BL/6 mice	40 μg course particulate matter/mL	Mice were placed 4 h/day or 5 days/week for 2 weeks in inhalation chamber	↑ Expression of TNF-α, IFN-γ, CXCL10, and IL-10 ↓ Expression of IL-5	[34]
Particulate matter	Urban particulate matter	C57BL/6 mice	200 μg	Gastric gavage	↑ ROS production ↑ Colonic epithelial cell death	[35]
Particulate matter	Atmosphericparticulate matter	Intestinal tissue	50–500 µg/cm^2^	1 week and 2 weeks	↓ ZO−1 and claudin−1 expression level	[36]
Lungs						
Nanoparticles	Nickel Oxide Nanoparticles	Rats	50 and 150 cm^2^ for 10 min	Intratracheally instilled 1 day and 4 weeks	-Narrowed alveolar ducts and alveoli ↑ Levels of neutrophils and cytokines -Induced pulmonary microbiome dysbiosis in the acute phase	[37]
Nano and microplastics	Polyethylene particles	BALB/c mice	10 mg/kg 1× daily for 1 week	Oral administration	↑ Percentage of Th17, Tregs, and Th2 cells ↓ Colon length ↑ Percentages of IL-4+, Foxp3+(Tregs), and IL-17 in CD4+T cells ↑ Secretion of IL-4 and IL-17 cytokines	[13]
Microplastics	PS-MPs	C57BL/6 mice	6.25 mg/kg PS-MPs	Intratracheally instilled 3× per week for 3 weeks	↑ Expression of collagen ↑ Fibrosis with increased exposure ↑ Oxidative stress	[38]
Ozone	Ozone exposure	C57BL/6 mice	1 ppm ozone for 1 h	Inhalation chamber 1×	↑ Barrier disruption and epithelial cell permeability	[39]
Ozone	Ozone exposure	C57BL/6 mice	1 ppm for the acute model and 1.5 ppm for the chronic model.	Acute phase: 1 h Chronic phase: 2 h, 2× per week for 6 weeks	Acute exposure induced the following: Disrupted tight junctions Desquamation of epithelial layer Chronic exposure to ozone particles remodeled the airway in mice using the following: ↑ Airspace density and diameter ↓ Number of airspaces ↓ Epithelium thickness	[14]
PAH	BaP	Sprague Dawley rats	0.01 mg/kg	Intratracheally instilled for 7 days	↑ Neutrophil recruitment ↑ Lung inflammation	[40]
PAH	Atmospheric PAHs	Human lung epithelial cell lines	Low dose	30 days	Chronic exposure: Induced DNA damage Altered cellular homeostasis and ↑ ROS production	[41]
Particulate matter	PM2.5	C57BL/6N mice	1.8, 5.4, and 16.2 mg/kg	Intratracheally instilled for 1 week	PM2.5 exposure led to the following: Infiltration of inflammatory cells, ↑ serum cytokine levels in the serum ↑ Lung microbiome diversity ↓ Relative abundance of proteobacteria post-exposure ↑ Number of goblet cells	[42]
Particulate matter	Various	18–64 years	Passive exposure	Average daily concentration	Exposure to air pollutants increased the risk for respiratory tract infections.	[43]
Circulatory system						
Heavy Metal	Cadmium	Cohort study	Passive exposure	Measured urinary cadmium concentration	May increase the incidence of ischemic stroke	[44]
Heavy Metal	Mixed metals	Ischemic stroke patients	Passive exposure	Fasting blood concentration within 48 h post-diagnosis	Higher plasma concentrations of aluminum, arsenic, and cadmium may increase the risk of ischemic stroke.	[45]
Dioxin	TCDD	Primary human aortic endothelial cell	0.1% TCDD	24 h	↑ Hypertension and endothelial dysfunction.	[46]
PAH	BaP	Sprague Dawley rats	0.01 mg/kg	Intratracheally instilled for 7 days	↑ Systolic and diastolic pressure and heart rate	[40]
PAH	BaP	Wistar rats	20 mg/kg for 4 and 8 weeks	Intra-peritoneal. injection	↑ Blood pressure after 8 weeks in vivo ↑ Vasoconstriction ex vivo	[47]
POPs	Passive exposure	Aged 70 yrs. in Sweden. 5-year follow-up	Passive exposure	Baseline plasma samples	PCB congeners, organochlorine pesticides, and octachlorodibenzo-p-dioxin showed increased risk of developing stroke in elderly population.	[48]
PCBs	Various PCBs	Endothelial cells from porcine pulmonary arteries	PCB 77, PCB 153, and PCB114	24 h	Exposure led to the development of atherosclerosis and endothelial barrier dysfunction. ↑ Albumin flux and oxidative stress	[49]

BaP, benzo-a-pyrene; CXCL10, interferon gamma-induced protein 10; DNA, deoxyribonucleic acid; Foxp3, forkhead box P3; IFN-γ, interferon-gamma; IL-4, interleukin-4; IL-5, interleukin-5; IL-6, interleukin-6; IL-10, interleukin-10; IL-17, interleukin-17; Ldlr−/−, LDL receptor deficient mouse; NF-κB, nuclear factor-κB; PAH, polycyclic aromatic hydrocarbon; PCB, polychlorinated biphenyls; PET, polyethylene terephthalate; PM2.5, particulate matter ≤ 2.5 μm; PS-MP, polystyrene microplastics; POP, persistent organic pollutant; ROS, reactive oxygen species; TCDD, 2,3,7,8-tetrachlorodibenzo-p-dioxin; Th2, T helper 2 cells; Th17, T helper 17 cells; TNF-α, tumor necrosis factor-alpha; Tregs, regulatory T cells; ZO-1, zona occludens 1. Upward and downward arrows indicate an increase or a decrease of the observed effect, respectively.

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
