# Peer review of "Exposure to Environmental Toxins: Potential Implications for Stroke Risk via the Gut– and Lung–Brain Axis"

_cells, 2024, doi:10.3390/cells13100803_

Round 1

Reviewer 1 Report

Comments and Suggestions for Authors

Dear Authors,

The article: 'Exposure to Environmental Toxins: An Emerging Stroke Risk via Regulation of The Gut-Lung-Brain Axis' presents a review that highlights the relationship between exposure to environmental toxins and increased risk of neurological disorders, particularly stroke. The study addresses a significant research gap by focusing on the impact of air pollution on stroke risk. This approach contributes to the broader understanding of environmental risk factors for brain diseases. However, it is important to note that there are some limitations in the review that need to be improved:

1. The article could benefit from further elaboration on the methodologies used to select and analyze evidence. To enhance the reliability of the findings, it is important to provide transparency regarding search strategies and inclusion criteria. Furthermore, it would be helpful to specify the type of review conducted (narrative or systematic).

2. It may be beneficial to consider condensing the sections into concise paragraphs with a clear sequence of information. For instance, it may be helpful to present experimental aspects and clinical/epidemiological study aspects separately. Furthermore, organizing pollutant categories could improve clarity. When selecting studies, it is recommended to prioritize those in which a cause-and-effect relationship has been established. It is advisable to base publications on works that provide hard data, proving that toxins directly contribute to causing stroke. It is important to rely less on logical inference and focus on far-reaching conclusions and hypotheses supported by evidence.

3. The individual sections contain many small paragraphs that can be logically combined. Subheadings can increase readability, but some subsections may be repetitive. For example, in section 3.2. The Gut-Brain-Axis and Environmental Toxins and 3.2.2. Environmental Toxins and the Gut Microbiota may be discussing the same subject matter. Please rethink subsections again.

4. A more detailed description below the figure is required.

5. The Authors note that Figure 1 was adapted from sources [180-182]. However, it is unclear whether permission was obtained from the original authors. It may be beneficial to address these issues for the sake of accuracy. Furthermore, the sources cited in the text are not included in the reference list [ref. 180-182]. Please complete and correct such inaccuracy.

I recommend addressing the mentioned points for improvement. With these refinements, the manuscript holds the potential for publication.

Author Response

We would like to thank all three reviewers for having taking time to carefully reviewed our manuscript and appreciate the relevant comments. We have addressed all the suggestions and comments (below and highlighted in blue in the manuscript). We hope we have now improved the manuscript. Our answers to all three reviewers can be find in the word file attached.

Reviewer 1:

  1. The article could benefit from further elaboration on the methodologies used to select and analyze evidence. To enhance the reliability of the findings, it is important to provide transparency regarding search strategies and inclusion criteria. Furthermore, it would be helpful to specify the type of review conducted (narrative or systematic).

Answer: Dear Reviewer, thank you very much for this suggestion, to enhance the transparency we have noted in the introduction section that this review article is a narrative review – page 2, line 70.

  1. It may be beneficial to consider condensing the sections into concise paragraphs with a clear sequence of information. For instance, it may be helpful to present experimental aspects and clinical/epidemiological study aspects separately. Furthermore, organizing pollutant categories could improve clarity. When selecting studies, it is recommended to prioritize those in which a cause-and-effect relationship has been established. It is advisable to base publications on works that provide hard data, proving that toxins directly contribute to causing stroke. It is important to rely less on logical inference and focus on far-reaching conclusions and hypotheses supported by evidence.

Answer: We agree that the sections can be more concise and better organized to improve readability. To better guide the reader, we have now added:

  • We re-organized the sections and a sentence describing the structure of the narrative review mentioning each environmental toxin categories that will be discussed (page 2, lines 94-97):

“In this section, we discuss pertinent environmental toxins and their metabolic pathways that lead to host toxicity, particularly in the context of stroke. The categories of toxins reviewed include: 1) Persistent Organic Pollutants (POPs), 2) Per- and Polyfluorinated Substances (PFAS), 3) Air Pollutants, and) Micro- and Nano plastics.“

  • Within each section, we separated more clearly experimental aspects from clinical studies.

  • We agree the selection of studies should be cause-and effect and therefore removed studies that do not bring substantial evidence to support this review.

  1. The individual sections contain many small paragraphs that can be logically combined. Subheadings can increase readability, but some subsections may be repetitive. For example, in section 3.2. The Gut-Brain-Axis and Environmental Toxins and 3.2.2. Environmental Toxins and the Gut Microbiota may be discussing the same subject matter. Please rethink subsections again.

Answer: We agree with your suggestion that adjustments to the organization and structuring would improve the readability and flow of the text. In Section 3.2, we have combined the Gut-Brain-Axis and The Gut Microbiota to reduce redundancy and improve organization. We also improved clarity and eliminate repetition in section 2.

  1. A more detailed description below the figure is required.

Answer: Indeed, the figure lacked significant information. We have significantly updated the figure and provided a more detailed description below.

  1. The Authors note that Figure 1 was adapted from sources [180-182]. However, it is unclear whether permission was obtained from the original authors. It may be beneficial to address these issues for the sake of accuracy. Furthermore, the sources cited in the text are not included in the reference list [ref.180-182]. Please complete and correct such inaccuracy.

Answer: Thank you raising the concern regarding permission and references. Indeed, this was a numbering error when copying the drafts to the required manuscript template. We have updated the figure and removed the citations. The reason we had the citations was because these publications were coming from our lab. To avoid any confusion with the need for copyrights we have adjusted the figure and no longer require citations or permissions. 

Reviewer 2 Report

Comments and Suggestions for Authors

This paper reviews the evidence for the association between environmental toxins as stroke risk factors by their actions in the gut and lungs. The authors describe how these enter the body by dietary intake, inhalation or dermal absorption, and by accumulating in the gut, lungs and central nervous system, affect host cellular responses and microbiome at gut and lung barrier sites,  interfere with cellular processes inducing neurotoxicity and altered immune response, increasing the risk of stroke. Lung and intestinal microbiota could be a significant intermediary in the influence of these chemical pollutants on neurological disorders. The ape reviews the role of environmental toxins and stroke risk - polycyclic aromatic hydrocarbons (PAHs), persistent organic pollutants (POPs), Per- and Polyfluorinated Substances (PFAS), Air Pollutants, Particulate Matter (PM), Nano- and Microplastics

Please attend to grammar issues eg line 132 – use ‘increase’ instead of ‘increasing’, line 157 - add ‘in’ in ‘accumulate lipid’, line 192, use ‘while’ instead of ‘whereas’, etc

Line 151 – should ‘2.1’ be ‘2.2’?

Line 175 – should ‘2.1’ be ‘2.3’?

Line 261 – should ‘2.5’ be ‘2.6’?

Table 1 – please add below the table a legend with explanations for the various abbreviations eg  ZO-1, TNF, IL, GIPS-MP, PCB, PPAR, NF, DNA, ROS, etc

Figure 1 - please add below the figure a legend with explanations for the various abbreviations eg  IL, TNF, NETs, ROS, BBB, TGF, etc. It it ‘comprised’ or ‘compromised’? Why s there a ‘mucus layer’ in the left arm? Are the authors meaning the gut? Isn’t there a mucus layer in lungs too, or is it not relevant?

A paragraph on the knowledge gaps would be helpful.

Comments on the Quality of English Language

Minor issues

Author Response

We would like to thank all three reviewers for having taking time to carefully reviewed our manuscript and appreciate the relevant comments. We have addressed all the suggestions and comments (below and highlighted in blue in the manuscript). We hope we have now improved the manuscript. Our answers to all three reviewers can be find in the word file attached.

Reviewer 2:

  1. Please attend to grammar issues eg line 132 – use ‘increase’ instead of ‘increasing’, line 157 - add ‘in’ in ‘accumulate lipid’, line192, use ‘while’ instead of ‘whereas’, etc

Answer: Dear Reviewer, thank you for your grammar edits. We have thoroughly checked to ensure all the language is consistent and have included your grammar suggestions.

  1. Line 151 – should ‘2.1’ be ‘2.2’? Line 175 – should ‘2.1’ be ‘2.3’? Line 175 – Should be ‘2.1’ be ‘2.3’?

Answer: Thank you for noticing we made a mistake in numbering subparagraphs. We have ensured the numbering is corrected throughout the document. Because we submitted a Word doc. file, this might be affected once again when opening the file using another Word version. We will review this carefully during editorial review.

  1. Table 1 – please add below the table a legend with explanations for the various abbreviations eg ZO-1, TNF, IL, GIPS-MP, PCB, PPAR, NF, DNA, ROS, etc

Answer: We agree with the reviewer that a table legend with explanations of the various abbreviations is missing. We have added a description and legend to improve the readability of the table.

  1. It it ‘comprised’ or ‘compromised’? Why s there a ‘mucus layer’ in the left arm? Are the authors meaning the gut? Isn’t there a mucus layer in lungs too, or is it not relevant?

Answer: Thank you for pointing out this error. Indeed, this zoom box is not supposed to be from the arm, but rather the gut. We have shifted the zoom box and made changes to the overall graphic to highlight that environmental toxins disrupt the gut. We also agree that the lungs do have a mucosal layer that is altered after toxin exposure. We have modified the figure to incorporate the reviewers’ comments.

  1. A paragraph on the knowledge gaps would be helpful

Answer: We agree that the point of a review is to highlight gaps in the literature and a perspective on outlook for future research. In the conclusion paragraph, we have added gaps in the literature stating on page 14, lines 503-513.

Reviewer 3 Report

Comments and Suggestions for Authors

The authors of the study examined recent evidence of how environmental toxins, absorbed along the digestive tract, or inhaled through the lungs, affect the host cellular response.

A.    The proficiency in the English language in this study is satisfactory.

B.      Title: I have the impression that the title doesn't fully correspond to what is written in the paper. The focus in the paper is not that much on stroke (at least in my opinion), but in the title, it is.

C.     Abstract: The abstract is well written.

D.     Introduction: The introduction is adequately written.

Line 116: “BaP leads to the induction of several deleterious pathways”- which one? This could be important information.

Lines 165 and 166: Could you elaborate more about congeners 138, 153 and 180?

2.1. Per- and Polyfluorinated Substances (PFAS): What signalling pathways or deleterious mechanisms PFAS trigger?

Lines 227-230: By which mechanisms does it trigger chronic neuroinflammation?

I'm not sure if the journal's propositions allow it, but Table 1 could have a slightly smaller font, which would make it more readable. Perhaps aligning it to the left could also help.

„Short-term exposure to PM10 reduced the tight junction protein occludin in human and rat alveolar epithelial cells [103]. In contrast exposure to PCB126, a dioxin-like compound increased the expression of the tight junction proteins occludin and claudin [104]. Jang et al., found that ozone exposure also significantly impacts the structural integrity of the lung's epithelial layer and immune response in the lungs. “- What is the conclusion for observed phenomena?

E.     Conclusion: It is well written.

Overall, this paper is extremely interesting to read and can provide useful information. However, I have the impression that everything is left undefined; everything is nicely written and organized, but there is a lack of more discussion about the mechanisms of how toxins actually affect the brain. In this version, everything is just listed (regarding the what can influence brain) without concrete evidence. I know it's a big task, but it would be nice if the authors could at least include a little bit about the mechanisms of actions that lead to the brain damage.

Comments on the Quality of English Language

Minor errors detected.

Author Response

We would like to thank all three reviewers for having taking time to carefully reviewed our manuscript and appreciate the relevant comments. We have addressed all the suggestions and comments (below and highlighted in blue in the manuscript). We hope we have now improved the manuscript. Our answers to all three reviewers can be find in the word file attached.

Reviewer 3:

  1. The proficiency in the English language in this study is satisfactory.

Answer: Dear Reviewer, thank you for defining the proficiency. We have thoroughly checked the grammar to improve above satisfactory level.

  1. Title: I have the impression that the title doesn't fully correspond to what is written in the paper.

The focus in the paper is not that much on stroke (at least in my opinion), but in the title, it is.

Answer: Dear reviewer, we appreciate your comment in regards to the title and the focus on the paper. We agree with the reviewer that there are so far limited studies showing the direct role of environmental toxins in stroke, but we believe our narrative review highlights this knowledge gap and address potential implication of environmental toxins for stroke. We have changed the title to “Exposure to Environmental Toxins: Potential Implications for Stroke Risk via The Gut- and Lung- Brain Axis”, to address the reviewer’s comment.

  1. Abstract: The abstract is well written.
  2. Introduction: The introduction is adequately written.

Answer: Thank you for the positive feedback regarding the abstract and introduction.

Line 116: “BaP leads to the induction of several deleterious pathways”- which one? This could be important information.

Answer: We agree the “deleterious pathways” should be mentioned. However, we decided to remove the sentence:  

“The biotransformation of BaP leads to the induction and inhibition of several deleterious pathway including the AhR and NF-κB pathway (PMID: 31103010 ), having mutagenic properties inducing abnormal DNA replication as wells as the generation of oxidative stress by the production of reactive oxygen species (ROS) (PMID: 16500718).”

Because this is not directly connected to the implication of these pathways modulated by BaP and stroke. It is very well known that ROS and NF-κB signaling contributes to stroke pathobiology but to reduce speculation and mention causality of toxins in stroke, we removed this sentence.

Lines 165 and 166: Could you elaborate more about congeners 138, 153 and 180?

Answer: Indeed, more information regarding these congeners would be helpful as these are all mid- and highly chlorinated congeners. We have included this information on page 4, lines 155-157.

2.1. Per- and Polyfluorinated Substances (PFAS): What signalling pathways or deleterious mechanisms PFAS trigger?

Answer: We agree that this information was not indicated. We have now added and elaborated on signaling pathways of PFAS in new section 2.2 (former section 2.3), page 4, lines 187-199. We hope this provides more insight to which signaling pathways are activated upon exposure.

Lines 227-230: By which mechanisms does it trigger chronic neuroinflammation?

Answer: Dear Reviewer, have added more information about what triggers chronic neuroinflammation in page 5, lines 227-229:

“Particulate matter can enter the circulation and can traffic to the brain acting as a proinflammatory stimulus and triggering chronic inflammation [74]. This sustained brain inflammation has been characterized as a consequence of continuous particulate matter exposure inducing ROS production, microglial activation, neuroinflammation, and neuronal damage [23,75–77].”

I'm not sure if the journal's propositions allow it, but Table 1 could have a slightly smaller font, which would make it more readable. Perhaps aligning it to the left could also help.

Answer: Thank you for noting how to improve the readability of the article. We will discuss formatting with the publishers.

„Short-term exposure to PM10 reduced the tight junction protein occludin in human and rat alveolar epithelial cells [103]. In contrast exposure to PCB126, a dioxin-like compound increased the expression of the tight junction proteins occludin and claudin [104]. Jang et al., found that ozone exposure also significantly impacts the structural integrity of the lung's epithelial layer and immune response in the lungs. “- What is the conclusion for observed phenomena?

Answer: We agree there is a lack of conclusion in this section. We have added the following conclusion on page 10, lines 304-308:

 “Overall, these findings highlight the complex interaction between environmental toxins and integrity of the epithelial barrier and immune regulation. While different environmental pollutants can have contrasting effects on epithelial barrier function and immune regulation, exposure may have implications for respiratory health and disease susceptibility.”

  1. Conclusion: It is well written.

Answer: Dear Reviewer, we appreciate your positive feedback regarding the conclusion. 

Overall, this paper is extremely interesting to read and can provide useful information. However, I have the impression that everything is left undefined; everything is nicely written and organized, but there is a lack of more discussion about the mechanisms of how toxins actually affect the brain. In this version, everything is just listed (regarding the what can influence brain) without concrete evidence. I know it's a big task, but it would be nice if the authors could at least include a little bit about the mechanisms of actions that lead to the brain damage.

Answer: Dear Reviewer, we are please to hear you found this paper interesting and useful. We have added more about the mechanisms throughout the paper. For instance, we added how BaP can accumulate in the brain and impair blood brain barrier in lines 137 to 140. According to reviewer 2’s comment, we also addressed how toxins (particulate matter) affect chronic inflammation, lines 227 to 229. Lines 342 to 345 highlights the impact of air pollution on stroke motor outcomes. The mechanism on how particulate matter enters the brain is elaborated on in lines 366 to 369. In addition, we have substantially improved figure 1 and added more details about the mechanisms of action of different toxins which have indicated to trigger brain damage.

Round 2

Reviewer 3 Report

Comments and Suggestions for Authors

The authors have answered all of my questions and the paper is now greatly improved. Therefore, it can be accepted for publication.